# Maternal Body Mass Index and Breastfeeding Non-Initiation and Cessation: A Quantitative Review of the Literature

**DOI:** 10.3390/nu12092684

**Published:** 2020-09-02

**Authors:** Kyoko Nomura, Sachiko Minamizono, Kengo Nagashima, Mariko Ono, Naomi Kitano

**Affiliations:** 1Department of Environmental Health Science and Public Health, Akita University Graduate School of Medicine, Akita 010-8543, Japan; sachikot@med.akita-u.ac.jp; 2Research Center for Medical and Health Data Science, The Institute of Statistical Mathematics, Tokyo 190-8562, Japan; nshi@ism.ac.jp; 3Teikyo University School of Medicine, Tokyo 173-8605, Japan; malinkopf@gmail.com; 4Research Center for Community Medicine and Department of Public Health, Wakayama Medical University School of Medicine, Wakayama 641-8509, Japan; naomiuk@wakayama-med.ac.jp

**Keywords:** breastfeeding non-initiation/cessation, exclusive/any breastfeeding, maternal BMI, meta-analysis

## Abstract

This study aims to investigate which maternal body mass index (BMI) categories are associated with the non-initiation or cessation of breastfeeding (BF) based on a quantitative review of the literature. We searched Ovid MEDLINE and EBSCO CINAHL for peer-reviewed articles published between 1946 (MEDLINE) or 1981 (CINAHL), and 2019. Selected studies were either cross-sectional or cohort studies, of healthy mothers and infants, that reported nutrition method (exclusive/full or any) and period (initiation/duration/cessation) of breastfeeding according to maternal BMI levels. Pairwise meta-analyses of 57 studies demonstrated that the pooled odds risks (OR) of not initiating BF among overweight and obese mothers compared to normal weight mothers were significant across 29 (OR 1.33, 95% confidence interval (CI), 1.15–1.54, I^2^ = 98%) and 26 studies (OR 1.61, 95% CI, 1.33–1.95, I^2^ = 99%), respectively; the pooled risks for BF cessation were inconsistent in overweight and obese mothers with substantial heterogeneity. However, we found that overweight mothers (*n* = 10, hazard ratio (HR) 1.16, 95% CI, 1.07–1.25; I^2^ = 23%) and obese mothers (*n* = 7, HR 1.45, 95% CI: 1.27–1.65; I^2^ = 44%) were both associated with an increased risk of not continuing any BF and exclusive BF, respectively. Overweight and obese mothers may be at increased risk of not initiating or the cessation of breastfeeding.

## 1. Introduction

Breastfeeding has overwhelming positive evidence for both infants and mothers. Breast milk consists of bioactive factors that play a role in immunological strengthening [1], carcinogenic development [2], neural and psychological benefits [3], and a possible protective factor for obesity in childhood [4]. In addition, recent literature demonstrates that children who are exclusively breastfed have lower infectious morbidity and mortality than those who are not breastfed, or partially breastfed [5]. Breastfeeding has a positive effect on women’s health in that it can prevent breast cancer [6], diabetes [7], and ovarian cancer [8]. According to a particular article published in the Lancet in 2016, the scaling up of breastfeeding can prevent an estimated 823,000 child deaths and 20,000 breast cancer deaths every year [9].

Despite such important features of breastfeeding practice, the global prevalence of breastfeeding at 12 months is lower than 20% in most high-income countries, such as the UK (<1%) [10], the USA (27%) [11], Norway (35%) [12], and Sweden (16%) [13]. A variety of factors determine breastfeeding non-initiation and cessation. These include biological factors such as maternal characteristics or delivery outcome, social and environmental factors, and even the perception of infant feeding preference of the parents and family [9]. Among these, there is also a growing set of studies from Western countries showing that maternal obesity might be associated with breastfeeding failure [14].

Nevertheless, these studies are still inconsistent due to the differences in ethnicity, population, body size classification, study design, sample size, risk measurement used, and confounder adjustment. Moreover, these studies hardly investigated the effect of being underweight on breastfeeding outcomes. Unlike Western countries where obesity is prevalent, underweight women of reproductive age in Asian countries [15] are seriously concerned about the health consequence of their infants (i.e., low weight infants) and the adverse effects on breastfeeding [16].

Hence, the purpose of this study was to investigate which maternal body mass index (BMI) categories (i.e., underweight or obesity) are associated with non-initiation or cessation of breastfeeding based on quantitative reviews.

## 2. Methods

The review followed the Preferred Reporting Items for Systematic Reviews and Meta-Analyses (PRISMA) [17].

### 2.1. Literature Search

A literature search was performed of the Ovid MEDLINE and EBSCO CINAHL databases from January 1946 and January 1981, respectively, through 8 June 2019 by using key terms (All Fields) of breastfeeding (“breastfeeding” or “breast feeding” or “breastfed” or “breastfeed” or “breast milk” or “human milk” or “lactation”) combined with type of breastfeeding (“exclusive” or “full” or “any”) and initiation or duration of breastfeeding (“initiation” or “duration” or “cessation”) and (“body mass index” or “BMI”). The search strategy is presented in Appendix A with the number of each search result according to the search term. The search process was initially confirmed by Osaka University Life Science Library and repeatedly updated by 3 authors (KN, NK, and SM) to ensure a reliable reference collection.

### 2.2. Selection Criteria

Studies meeting the following criteria qualified for inclusion in our meta-analysis: (i) the study was published as original research in any language; (ii) the study did not merely report proportions or differences, but a risk (i.e., odds/risk/hazard ratio; OR, RR, HR, respectively) of breastfeeding initiation/duration/cessation among overweight/obese and/or underweight women compared to normal weight women; (iii) breastfeeding was inclusive of any breastfeeding or/and exclusive breastfeeding (i.e., if infants were given no solid food or juice/water) that included full breastfeeding; (iv) breastfeeding cessation included disrupted lactation, undesired weaning, or delayed onset of lactogenesis; and (v) the study investigated maternal BMI in which the categories were clearly presented with a value range, or defined according to guidelines such as the World Health Organization (WHO) classification [18], the WHO Asian-specific pre-pregnancy body mass index [19], and the Institute of Medicine [20]. The WHO [18] classifies BMI into the following categories: underweight (<18.5 kg/m^2^), normal weight (18.5–24.9 kg/m^2^), overweight (25.0–29.9 kg/m^2^), and obese (≥30.0 kg/m^2^). The Asian-specific classification differs slightly from the original and defines overweight (23.0–24.9 kg/m^2^), obese I (25.0–29.9 kg/m^2^), and obese II (≥30 kg/m^2^). The Institute of Medicine [20] classifies BMI into underweight (<19.1 kg/m^2^), normal weight (19.1–26.0 kg/m^2^), overweight (26.1–29.0 kg/m^2^), and obese (≥29.0 kg/m^2^).

We excluded studies if: (i) the sample size of BMI categories was not provided; (ii) they investigated BMI but did not report BMI values and ranges; (iii) they investigated “intention” of breastfeeding practice only or did not investigate the actual status of breastfeeding; (iv) they investigated the relationship of diabetic mothers, because these mothers are more likely to modify their BMI by treatment or genetic reasons; (v) they investigated the risk of maternal BMI as a continuous variable (i.e., risk per one unit increase of pre-pregnancy BMI); (vi) the study had the same sample population of a study that had already been included in our meta-analyses; and (vii) the study had particularly focused on the very obese (i.e., obese class II, III) because such studies were too few to be included in the meta-analyses. Among these, there were 10 studies [21,22,23,24,25,26,27,28,29,30] which used continuous BMI, as provided in the Appendix A.

### 2.3. Data Extraction

After screening the titles and abstracts of the retrieved papers, two investigators (KN and NK) initially and independently read the relevant papers that passed the first stage of selection. Then, SM further reviewed all relevant papers, assessed eligibility, and resolved any disagreements if necessary. The extracted data included author (year), participants, country, ethnicity, BMI classification, the timing of BMI measurement, breastfeeding (BF) initiation/cessation, timing of risk assessment used for meta-analyses, BF prevalence, measure of association, and adjusted variables to estimate. Then, one investigator (KN) created a datasheet for analyses and computerized the relevant information, which was double checked by the statistician (Kengo Nagashima).

### 2.4. Outcome of Breastfeeding

The outcome of interest in our meta-analyses was any or exclusive breastfeeding (ABF and EBF, respectively) in relation to non-initiation or cessation. According to the definition proposed by the WHO [31], children were considered to be exclusively breastfed if they were given no other food, drink, added nonhuman milk or infant formula, except breast milk. ABF was considered if the child received liquids like water-based drinks, fruit juice, and ritual fluids, other than breast milk [31], or if the study did not clearly describe exclusive or any. We collected either ORs or HRs of BF non-initiation or cessation and, if the study provided risks of BF initiation or continuation, we recalculated the risk of BF non-initiation or cessation by taking the inverse of the risk of the BF status to align the direction of the risk. To estimate the pooled effect size of the risk measures, we used risk ratios in multivariable analyses rather than unadjusted models.

### 2.5. Quality Assessment

We rated the methodological quality of the studies according to a modified version of the criteria provided by the Newcastle–Ottawa scale for cohort studies [32] because our included studies were prospective, retrospective cohort, or cross-sectional studies. The Newcastle–Ottawa Scale uses a star system (with a maximum of 9 stars) to evaluate a study in 3 domains: selection of participants (4 items; Representativeness of the Exposed Cohort, Selection of the Non-Exposed Cohort, Ascertainment of Exposure, Pre-Pregnancy BMI Measurement), comparability of study groups (2 items; Control for Confounders), and the ascertainment of outcomes of interest (3 items; Assessment of Breastfeeding, Was Follow-Up Long Enough for Outcomes to Occur?, Adequacy of Follow-Up of Cohorts).

Representativeness of the Exposed Cohort earns one star if the representativeness of exposed individuals in the particular community is described or if the sample size is larger than 1000. Selection of the Non-Exposed Cohort earns one star if only the normal weight group was defined as a reference group. Ascertainment of Exposure earns one star if maternal weight and height were measured and not self-reported to calculate maternal BMI. Pre-Pregnancy BMI Measurement earns one star if the study clearly stated a BMI measurement in the earlier timing of pregnancy. This indicates that the study is prospective and breastfeeding was not present at the start of the study.

A maximum of 2 points can be given for comparability. If a study controls at least 2 variables, including maternal age, gestational week and delivery mode to calculate a risk of BF non-initiation or cessation, it will receive one point. If the study also controls for other confounding factors (i.e., socio-economic factors, psychological factors, and social support), it will receive another point.

The assessment of breastfeeding earns one star if initiation or continuation of the particular type of breastfeeding (i.e., exclusive, full, dominant, or any breastfeeding) is clearly defined. “Was Follow-Up Long Enough for Outcomes to Occur?” earns one star if the study calculated the risk estimates at a minimum of one month for initiation and 6 months for continuation, because the WHO recommends 6 months of breastfeeding. Adequacy of Follow-Up of Cohorts earns one star if the study was a cohort study and 70% of the initially enrolled sample was followed up, and N/A if the study was a cross-sectional study.

With a maximum of 9 stars, we judged studies that received a score of 9 stars to be at low risk of bias, studies that received 7 (for cross-sectional studies) or 8 stars (for cohort studies) to be at medium risk, and those that received 6 (for cross-sectional studies) or 7 stars (for cohort studies) (or fewer) to be at high risk of bias.

### 2.6. Data Analysis

Pairwise meta-analyses were performed to compare the estimate of BF non-initiation, and ABF or EBF cessation between each pre-pregnancy BMI category (i.e., underweight, normal weight as reference, overweight, and obesity). Pooled ORs and HRs and 95% confidence intervals (CIs) of being underweight, overweight, or obese on an effect of (i.e., above 1) breastfeeding non-initiation/cessation and were estimated using random-effects models with the Sidik–Jonkman estimator [33]. Subgroup analyses were conducted according to BF (ABF) or EBF, the timing of the risk assessment, ethnicity, and quality score (≤5 vs. ≥6). For a study reporting only subgroup results, we first estimated the combined ORs or HRs within the study by using random-effects models, and then combined these with other findings from the selected study [34]. Meta-regression analyses were also performed to examine the impact of categorical variables on study effect size using regression-based techniques. Heterogeneity across individual studies by Cochran’s Q (a measure of weighted squared deviations), the I^2^ statistics (the ratio of true heterogeneity to total observed variation), and Tau^2^ (the variance of the true effect sizes) were estimated. Tau^2^ can be viewed as a point estimate of the among-study variance of true effects, while I^2^ would be a measure of inconsistency and the proportion of variability in the point estimates [35]. For Cochran’s Q test, the P value indicating significant heterogeneity was set at less than 0.10 [36]. For I^2^, values less than 25%, from 25% to 50%, and more than 50% indicated modest, moderate and substantial heterogeneity, respectively. To assess publication bias, we constructed funnel plots and tested the asymmetry using Egger’s and Begg’s test. If Begg’s test was significant, we further constructed a trim and fill funnel plot by imputing missing studies to adjust for publication bias.

All the analyses were conducted in the statistical software package STATA version 16. *p* < 0.05 was considered statistically significant, unless otherwise specified.

## 3. Results

### 3.1. Search Result and Selection of Studies

Our literature search was performed on MEDLINE and CINAHL and 427 publications were identified. Of these, 122 were duplicated and excluded. We initially screened 305 papers by title and abstract search and excluded 230 papers. After adding 34 articles identified from manual searches and reviews of bibliographies of relevant studies, we critically reviewed 109 articles. After excluding 14 studies with different topics, we reviewed 95 articles for data extraction. We further excluded another 38 articles due to extreme obesity (*n* = 2) [37,38], pre-eclampsia (*n* = 1) [39], duplicate data sources (*n* = 1) [40], gestational diabetes mellitus (GDM) (*n* = 4) [41,42,43,44], inability to integrate (*n* = 20) [45,46,47,48,49,50,51,52,53,54,55,56,57,58,59,60,61,62,63,64] and continuous BMI (*n* = 10) [21,22,23,24,25,26,27,28,29,30], resulting in 57 articles that were included in the meta-analyses (Figure 1).

### 3.2. Study Characterisics

The characteristics of the 57 included articles are presented in Table 1. Among 57 studies, 43 were published in the US and European countries [65,66,67,68,69,70,71,72,73,74,75,76,77,78,79,80,81,82,83,84,85,86,87,88,89,90,91,92,93,94,95,96,97,98,99,100,101,102,103,104,105,106,107], eight in Australia [60,108,109,110,111,112,113,114], three in China and Japan [16,115,116], two in Kuwait and Iran [117,118] and one in Brazil [119]. Except for one study [65], all were published after the year 2000, of which 15 [60,66,67,68,69,70,71,72,73,74,75,108,109,110,111] were published between 2000 and 2009, and 41 studies were published between 2010 and 2019. The majority of the included studies reported adjusted risk of BF non-initiation or cessation, except for three studies [80,113,117]. Twenty-nine studies [16,60,65,66,67,70,73,74,76,77,78,79,83,86,88,89,96,97,98,99,104,105,106,107,110,111,112,114,118] investigated BF initiation by a logistic regression model. Forty studies [67,68,69,70,71,72,73,75,76,80,81,82,84,85,87,90,91,92,93,94,95,96,97,100,101,102,103,104,106,108,109,110,111,113,114,115,116,117,118,119] investigated BF cessation by either a logistic regression model or a Cox proportional hazard model or both, of which 14 studies [67,69,70,71,75,81,82,87,92,94,109,110,111,117] investigated BF continuation for six months or longer.

The quality assessment of each study according to the Newcastle–Ottawa scale [32] is illustrated, together with the number of stars awarded, in Appendix A. The overall quality score ranged from three to eight, with a mean score ± standard deviation of 4.8 ± 1.5. Among the 57 studies investigated, there were no studies that received a score of nine stars to be at low risk of bias. Instead, eight studies [71,96,97,101,110,111,115,119] received seven/eight stars and were considered to have a medium risk of bias, and the remaining 49 studies were awarded six or fewer stars. For maternal BMI, the majority of the studies investigated BMI before pregnancy but seven studies [84,100,101,103,108,109,111] investigated postpartum BMI. In addition, most of the studies collected BMI information by self-reporting of mothers and the timing of BMI measurement was not always clearly described.

### 3.3. Breastfeeding Non-Initiation

Table 2 shows the pooled risk for breastfeeding non-initiation with 95% CI, heterogeneity, and publication bias and Figure 2 shows a forest plot for BF non-initiation according to BMI categories. The OR for BF non-initiation was 1.03 (95% CI: 0.85–1.23; I^2^ = 88%, *p* for Q < 0.001, Tau^2^ = 0.08) among 14 studies [16,70,73,76,83,86,88,96,97,98,99,104,105,112] in underweight mothers. Although Begg’s test was non-significant, indicating that publication bias is less likely, a large I^2^ and significance of Q test indicated substantial heterogeneity existed among the studies. Our subgroup analyses with high quality score, BF type (ABF or EBF), timing of measurement (one month or no report), and ethnicity all showed insignificant results of the effect of being underweight on BF non-initiation (data not shown). The OR of BF non-initiation in overweight and obese mothers was 1.33 (95% CI: 1.15–1.54; I^2^ = 98%, *p* for Q < 0.001, Tau^2^ = 0.12) among 29 studies [16,60,65,66,67,70,73,74,76,77,78,79,83,86,88,89,96,97,98,99,104,105,106,107,110,111,112,114,118] and 1.61 (95% CI: 1.33–1.95; I^2^ = 99%, *p* for Q < 0.001, Tau^2^ = 0.20) among 26 studies [16,60,65,66,67,70,73,74,76,77,79,83,86,88,89,96,97,98,99,104,105,107,110,111,112,118], respectively. As the *p* value for Begg’s tests in both overweight and obese mothers was less than 0.05, indicating publication bias, we additionally performed a trim and fill funnel plot by imputing missing studies. We then observed the significance for obese (OR 1.32, 95% CI: 1.08–1.61) but not for overweight women (OR 1.13, 95% CI: 0.98–1.30).

### 3.4. Any Breastfeeding Cessation 

Table 3 shows the pooled risk for ABF cessation with 95% CI, heterogeneity, and publication bias and Figure 3 shows a forest plot for ABF cessation according to BMI categories. For studies that investigated underweight mothers, both the overall HR of four studies [70,76,102,115] (1.28, 95% CI: 0.71–2.31; I^2^ = 95%, *p* for Q = 0.024, Tau^2^ = 0.32) and the OR of 10 studies [69,71,73,92,96,97,101,103,109,116] (0.91, 95% CI: 0.64–1.28; I^2^ = 86%, *p* for Q = 0.002, Tau^2^ = 0.19) were insignificant. For studies that investigated overweight mothers, overall HR was significant (1.16, 95% CI: 1.07–1.25; I^2^ = 23%, *p* for Q = 0.913, Tau^2^ < 0.01) among 10 studies [67,70,76,81,82,84,102,110,111,115], indicating that overweight mothers had an increased risk of not continuing ABF compared to normal weight mothers. The significance was consistently observed in subgroup analyses of high quality scores ≥6 (*n* = 3 [67,110,116]; HR, 1.14, 95% CI: 1.03–1.26; I^2^ = 14%, *p* for Q = 0.620, Tau^2^ < 0.01), the six month assessment (*n* = 4 [67,80,81,102]; HR 1.22, 95% CI: 1.02–1.46; I^2^ = 29%, *p* for Q = 0.440, Tau^2^ < 0.01; data not shown), and white ethnicity (*n* = 7 [70,81,82,84,102,110,111]; HR 1.19, 95% CI: 1.09–1.29; I^2^ = 14%, *p* for Q = 0.910, Tau^2^ < 0.01; data not shown), confirming that overweight mothers had an increased risk of not continuing BF practices compared to normal weight mothers. On the other hand, the overall OR of being overweight on BF cessation was insignificant among 16 studies [69,71,72,73,91,92,93,96,97,101,103,109,110,111,116,117] with substantial heterogeneity (I^2^ 89%, *p* for Q <0.001, Tau^2^ = 0.07). For studies that investigated obese mothers, overall HR was significant (HR 1.43, 95% CI: 1.22–1.69) among 10 studies [67,68,70,76,82,84,94,102,108,115] with substantial heterogeneity (I^2^ 68%, *p* for Q = 0.081, Tau^2^ = 0.04). Such significance was also observed in the subgroup analyses including the six month assessment (*n* = 4 [67,82,94,102]; HR 1.66, 95% CI: 1.19–2.31; I^2^ = 67%, *p* for Q = 0.102, Tau^2^ = 0.08) and white ethnicity (*n* = 7 [68,70,82,84,94,102,108]; HR 1.19, 95% CI: 1.09–1.29; I^2^ = 14%, *p* for Q = 0.907, Tau^2^ < 0.01). The overall OR of being obese was significant (OR 1.47, 95% CI: 1.16–1.86) among 18 studies [71,72,73,75,80,85,91,92,93,96,97,101,103,109,110,111,113,117] but heterogeneity was substantial (I^2^ = 93%, *p* for Q < 0.001,Tau^2^ = 0.20). The subgroup analyses showed that the pooled risk among seven studies with a higher quality score [71,73,96,97,101,102,103,104,105,106,107,108,109,110,111] (OR 1.71, 95% CI: 1.34–2.18; I^2^ = 85%, *p* for Q < 0.001, Tau^2^ = 0.06), and the risk among 16 studies with white ethnicity [71,73,75,80,85,91,92,93,96,97,101,103,109,110,111,113] (OR 1.49, 95% CI: 1.14–1.95; I^2^ = 94%, *p* for Q < 0.001, Tau^2^ = 0.23; data not shown) were both significant, but that among 12 studies in the six month assessment [71,75,91,93,96,97,101,109,110,111,113,117] was insignificant (OR 1.37, 95% CI: 0.99–1.90; I^2^ = 96%, *p* for Q < 0.001, Tau^2^ = 0.29).

### 3.5. Exclusive Breastfeeding Cessation

Table 4 shows the pooled risk for not continuing EBF with 95% CI, heterogeneity, and publication bias and Figure 4 shows a forest plot for EBF cessation risk according to BMI categories. For the effect of being underweight on EBF cessation, both the overall HR and OR did not reach significance. For the effect of being overweight, although the overall HR was insignificant, the overall OR reached significance for overweight women having an increased risk of not continuing EBF compared to normal weight women (*n* = 12 [71,90,95,96,97,100,103,104,106,114,115,117]; OR 1.37, 95% CI: 1.10–1.72; I^2^ = 96%, *p* for Q < 0.001, Tau^2^ = 0.13). As Begg’s test was significant in overweight mothers, indicating publication bias, we additionally performed a trim-and-fill analysis by imputing missing studies and then observed insignificance (OR 1.16, 95% CI: 0.97–1.37). The significance was observed across subgroup analyses of quality score ≥6 (*n* = 7 [71,90,95,96,97,104,115]; OR 1.15, 95% CI: 1.08–1.24; I^2^ = 48%, *p* for Q = 0.059, Tau^2^ < 0.01) and white ethnicity (*n* = 9 [71,95,96,97,100,103,104,106,114]; OR = 1.51, 95% CI: 1.13–2.02; I^2^ = 93%, *p* for Q < 0.001, Tau^2^ = 0.16; data not shown) but not in a subgroup analysis of six month timing of measurement (*n* = 5 [71,97,106,115,117]; OR 1.45, 95% CI: 0.81–2.59; I^2^ = 98%, *p* for Q < 0.001, Tau^2^ < 0.01; data not shown). For the effect of obesity, the pooled risk was significant among seven studies [67,70,87,93,94,102,119] with HR (1.45, 95% CI: 1.27–1.65; I^2^ = 44%, *p* for Q = 0.455, Tau^2^ = 0.01) and insignificant among 11 studies [71,90,95,96,97,100,103,104,115,117,118] with OR (1.32, 95% CI: 0.97–1.79; I^2^ = 96%, *p* for Q < 0.001, Tau^2^ = 0.21).

## 4. Discussion

This study investigated which maternal BMI categories are associated with BF non-initiation or cessation using a quantitative review. The effects of being underweight on BF non-initiation and cessation were all insignificant irrespective of the types of BF practice (i.e., ABF or EBF), or risk measurement (i.e., HR or OR). The pooled risks of non-initiation among those who were overweight and obese were both significant, although there was substantial heterogeneity observed. The pooled risk for ABF and EBF cessation in overweight and obese mothers was inconsistent with substantial heterogeneity. However, the hazard risks of ABF cessation in overweight mothers and EBF cessation in obese mothers were both significant with less heterogeneity (i.e., I^2^ = 23% and 44%, respectively). This quantitative review suggests that overweight and obese mothers may have an associated increased risk of BF non-initiation or cessation.

There are two previous meta-analysis studies [14,120] that investigated an association between maternal weight status and BF non-initiation and cessation which agree that obesity may be associated with an increased risk of BF non-initiation or cessation compared to normal weight mothers. However, their study hypotheses differ from ours; one study [120] not only included maternal weight status, but diet and supplement use as determinants of breastfeeding and complementary feeding; the other [14] investigated BF cessation associated with maternal BMI and gestational weight gain. The study conducted by Huang et al. [14] is similar to ours, but does not include cross-sectional studies, and thus includes only 30 studies. The exclusion of these studies may overlook the true association of interest because most studies that focused on breastfeeding initiation are cross-sectional studies. Hence, the advantages of our study are that, first, the numbers of studies included in our meta-analyses are much larger in our study than in Huang et al. We included 57 studies (initiation *n* = 29, ABF cessation *n* = 31, EBF cessation *n* = 20) vs. 30 in Huang et al. (initiation *n* = 21, ABF cessation *n* = 15, EBF cessation *n* = 17). Second, we also performed a comprehensive literature review including a large number of references (i.e., 130 in our study vs. 54 in Huang et al.). For example, our study encompassed an extended literature search not only limited to the studies included in the meta-analyses, but incorporating those studies that were excluded into Appendix A. The description of studies excluded from the meta-analyses is important to reproduce a systematic review for a third person. Second, by performing pooled analyses according to odds or hazard ratios, the heterogeneity of some meta-analyses is less observed, which makes it clearer to present the true association of interest. The discrepancy is that Huang et al. reported underweight mothers to be less likely to initiate BF compared to normal weight mothers (*n* = 14, RR 1.28, 95% CI:1.11–1.48). However, we did not observe such a negative impact among underweight women; both studies observed substantial heterogeneity across the studies included in the meta-analyses. In this regard, the effect of the mother being underweight on breastfeeding still requires further evidence.

There are several reasons for the substantial heterogeneity across the studies. First, there were only two studies awarded the maximum number of seven stars for quality, and the other studies were awarded six stars or fewer, which indicates the majority of studies included in these meta-analyses had, at least, a moderate risk of bias. The reason for the lower quality scores across studies includes the nondifferential misclassification or recall bias by self-reporting of maternal weight. In addition, the majority of the reviewed studies failed to provide follow-up rates that were long enough to assess breastfeeding duration. Indeed, few studies in which mothers continued breastfeeding up to six months (as recommended by the WHO and UNICEF) remained for analyses. Second, we excluded 10 studies [21,22,23,24,25,26,27,28,29,30] that provided continuous risk estimates because risk with a one unit increase in BMI is exponential, and thus difficult to interpret by combining the different sources of populations. Indeed, the 10 studies [21,22,23,24,25,26,27,28,29,30] vary across countries, BF types, and the length of the follow-up period. Nine studies [21,22,23,24,25,26,27,28,29] were reported from Western countries and three investigated initiation [23,25,27]. Among the seven studies [21,22,23,24,26,29,30] that assessed the risk of timing at six months after delivery, only two [27,30] reported that the BF significantly decreased by a unit increase of BMI. Furthermore, we also attempted to collect mean BMI according to BF status in order to perform a quantitative assessment, but only two studies [25,29] provided such data. These two studies also reported the risks at different points in time (i.e., 18 months [25] and six months [29]) and thus we did not include these. Third, although we were able to assess the impact of being underweight on breastfeeding practice, because the studies in this area of research focus on obesity, the numbers of underweight studies included in the meta-analyses were few, especially for EBF cessation. Fourth, the WHO encourages EBF, especially in developing countries where maternal biological characteristics and socio-economic factors are significantly different from Western countries. In this regard, although there was no publication bias observed, the number of studies published from developing countries were still few, and thus the generalizability of the present meta-analyses may be limited.

Previously, several potential mechanisms have been reported. First, overweight or obese mothers have large heavy breasts, which may physiologically interfere with infant latching or adversely affect lactation [88]. Rasmussen and Kjolhed [121] demonstrated that excess adiposity in obese women contributes to dysregulation of the hypothalamic–pituitary–gonadal axis. In those women, the prolactin response to the baby’s suckle is low with a delayed onset of milk production [122]. Some evidence from research in dairy cows and mice suggests that obesity in early life may negatively influence breast glandular development [123]. Obese mice also show a reduction in the milk proteins β-casein, whey acidic protein, and α-lactalbumin, which are essential for milk production [124]. The breastmilk of normal weight women contains a higher concentration of medium-chain fatty acids; on the contrary, the milk of obese women is characterized by the presence of longer-chain fatty acids. These are more difficult to digest, especially in newborns with immature gastrointestinal systems, thus favoring supplementation with formula and/or solid food [125]. Another study showed that overweight women have a negative physiological pattern [126] influencing “maternal–fetal attachment”. Overweight and obese women who are less likely to intend to breastfeed [127], had less confidence in sufficient milk supply and lower body shape satisfaction [128]. They often reported postpartum depression [129], and therefore bottle-feeding is more precocious. However, this association may be confounded by biological and social factors [130]. Finally, breastfeeding practice may be influenced by sociocultural factors. As our Begg’s test indicates that publication bias may exist in our subgroup analyses, the majority of the reviewed studies were reported from developed countries that would be a great source of heterogeneity.

## 5. Conclusions

In summary, despite substantial heterogeneity across the reviewed studies, our quantitative review suggests that overweight and obese mothers are associated with breastfeeding non-initiation and cessation. Breastfeeding determinants are multifaceted, but among the relevant factors weight control is important to contribute to breastfeeding practice. This should be kept in mind for all relevant health care workers who are responsible for catering to women of reproductive age.

## Figures and Tables

**Figure 1 nutrients-12-02684-f001:**
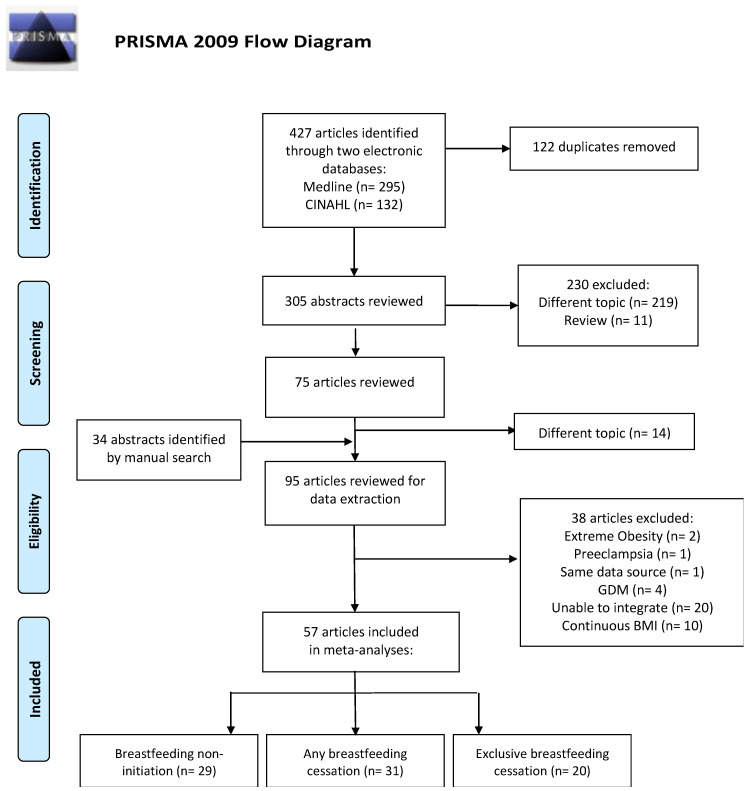
Selection of studies published from 1946 (MEDLINE) and 1981 (CINAHL) to 2019 and included in a meta-analysis of maternal BMI and breastfeeding practice. GDM-gestational diabetes mellitus.

**Figure 2 nutrients-12-02684-f002:**
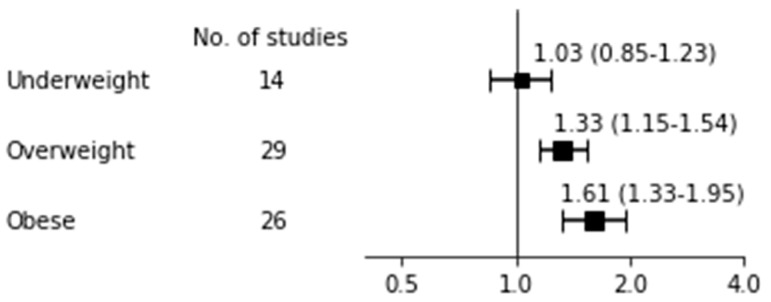
Breastfeeding non-initiation.

**Figure 3 nutrients-12-02684-f003:**
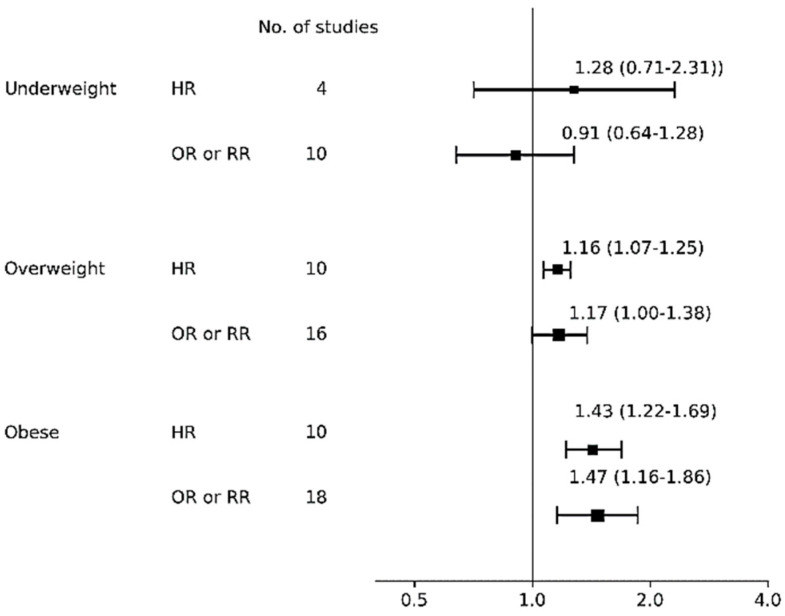
Forest plot for any breastfeeding cessation risk. Abbreviations: HR, hazard ratio; OR odds ratio; RR, risk ratio.

**Figure 4 nutrients-12-02684-f004:**
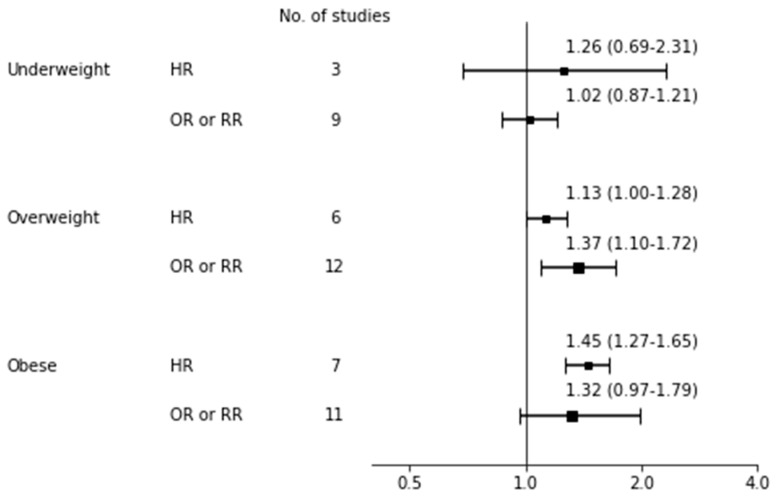
Forest plot for exclusive breastfeeding cessation risk. bbreviations: HR, hazard ratio; OR odds ratio; RR, risk ratio.

**Table 1 nutrients-12-02684-t001:** The characteristics of the 57 included articles.

Author (Year)	Participants	Country	Ethnicity	BMI	BF	Initiation or Cessation for Meta-Analyses	Timing of Risk Ratio Included in Meta-Analyses		BF Proportion (if Relevant Timing Available)	Measure of Association	Adjusted Variables to Estimate Risk of Breastfeeding Non-Initiation or Cessation	Quality
							Initiation	Cessation				
Hilson (1997) [65]	810	US	C 99%	<26.1 (ref), 26.1–29, ≥29.1	ABF/EBF	Initiation	Discharge	-	75% (at delivery)	OR/HR	P, GW, BW, Age, Edu, Participation in WIC, PCAP, DML, DM, Smk	4
Donath and Amir (2000) [108]	1991 mothers who had children under age 4 years	Australia	-	<25 (ref), 25–30, ≥30.1	ABF	Cessation	-	N/A	87% (ever breastfed)	HR	Age, Edu, Smk, Mar, SC, Housing	4
Sebire (2001) [66]	287,213	UK	W 72%	20–25 (ref), 25–30, ≥30.1	ABF	Initiation	Discharge	-	-	OR	R/Eth, P, Age, HTN, DM	3
Kugyelka (2004) [67]	B 263, H 235	US	B 53%, H 47%	<19.1–26.0 (ref), 26.1–29, ≥29.1	ABF/EBF	Initiation/Cessation	Discharge	6 months	Black 45%, Hispanic 59% (ever breastfed in hospital)	OR/HR	Age, Edu, P, GW, BW, Smk, DLM	6
Hilson (2004) [68]	151	US	W	<26.0 (ref), 26.1–29, ≥29.1	ABF	Cessation	-	N/A	-	HR	PDB, RTW, Lower satisfaction of appearance, Greater indifference towards breastfeeding	4
Grjibovski (2005) [69]	1078	Russia	Russian	Underweight, Normal (ref), Overweight	ABF	Cessation	-	12 months	96% (1 month) 18% (12 months)	OR	Age, Edu, O, Mar, DLM, GW, BW, InfSex, FTIPC, P	6
Forster (2006) [109]	764	Australia	Aus 70%	<20, 20–25 (ref), 26–29, ≥29.1	ABF	Cessation	-	6 months	55% (6 moths)	OR	Age, PIFI, Desire, PH, Smk, RH, ROB, PCAP, AOD, RP	4
Hilson (2006) [70]	876 who gained weight within the IOM recommendation	US	W	<19.8, 19.8–26.0 (ref), 26.1–29.0, ≥29.1	ABF	Initiation/Cessation	Discharge	24 months	91% (initiated)	OR/HR	Edu, Smk, Age, P, Participation in WIC/PCAP, DLM	5
Oddy (2006) [110]	1803	Australia	C 86%	<25 (ref), 25–29.9, ≥30	ABF	Initiation/Cessation	1 month	2, 4, and 6 months	91% (1 month), 74% (2 months), 59% (4 months), 49% (6 months)	OR/HR	InfSex, GW, BW, Edu, Age, Eth, P, Smk, DLM, PB, Age solids introduced	7
Scott (2006) [60]	556	Australia	W	<25 (ref), 25–29.9, ≥30	ABF/EBF	Initiation	Discharge	-	ABF 94%, EBF 77%	OR	Age, MS, O (mother, father), ROB, PCAP, DLM, ADM, FP (father, grandmother), PH (grandmother), Timing, Smk, IFS (mother)	4
Baker (2007) [71]	37,459	Denmark	Danish	<18.5, 18.5–24.9 (ref), 25.0–29.9, 30.0–34.9,	ABF/EBF	Cessation	-	6 months	65% (6 months)	IRR/OR	Age, GWG, POC, O, P, Smk, DLM, PA, InfSex	8
Jain (2007) [72]	7661	US	Multiethnicities	19.8–26.0 (ref), 26.1–29.0, ≥29.1	ABF	Cessation	-	10 weeks	47% (10 weeks)	OR	BMI, GWG, Eth, Age, Edu, P	5
Donath and Amir (2008) [111]	3075	Australia	-	20–24.9 (ref), 25–29, ≥29.1	ABF	Initiation/Cessation	1 week	6 months	88% (1 week), 57% (6 months)	OR	Age, Edu, Smk, ADM, SED, DLM	7
Manios (2009) [73]	1897 children aged 12 to 60 months	Greek	Greek	<19.8, 19.8–26.0 (ref), 26.0–29.0, ≥29.1	ABF	Initiation/Cessation	-	N/A	-	OR	BW, GW, P, Age, Edu, Smk	6
Fairlie (2009) [74]	1436	US	W 74%	20–24.9 (ref), 25–29.9, ≥30	ABF	Initiation	-	-	86%	OR	Age, Eth, GW, DLM, Inc, Edu, ROB, POC	5
Kehler (2009) [75]	780	Canada	Can 83%	<18.5, 18.5–24.9 (ref), 25.0–29.9, ≥30	ABF	Cessation	-	6 months	72% (6 months)	OR	RTW or intending to within first postpartum year, AOD	4
Liu (2010) [76]	3517 (White) 2840 (Black)	US	W 55%, B 45%	<18.5,18.5–24.9 (ref), 25–29.9, 30–34.9	ABF	Initiation/Cessation	After delivery	10 weeks	W 67%, B 41% (after delivery); W 55%, B 42% (10 weeks)	OR/HR	Age, Edu, Mar, PB, Smk, BW, P, FTIPC	4
Kitsantas (2010) [77]	10,700	US	W 60%	<18.5,18.5–24.9 (ref), 25–29.9, 30–34.9	ABF	Initiation	-	N/A	69%	OR	Age, Mar, Eth, P, Smk, Alc, DLM, InfSex, GW, BW	5
Biro (2011) [112]	3352	Australia	98% Non-aboriginal	<20, 20–24.9 (ref), 25–29.9, ≥30	ABF	Initiation	After delivery	-	82% (after delivery)	OR	P, Age, HC, Edu, Mar, ROB, Smk, BW, ADM, BFHI accreditation, MOI, Analgesia, DML, BW, ADM	4
Mehta (2011) [78]	688	US	W 77%	≤26 (ref), ≥26.1	ABF	Initiation	3 months	-	91%	OR	R/Eth, Edu, Mar, PS	3
Tenfelde (2011) [79]	235	US	Mexican 42%	<18.5, 18.5–24.9 (ref), 25.0–29.9, ≥30	EBF	Initiation	1 month	-	23%	OR	FTIPC, RTW, PIFI,	2
Leonard (2011) [80]	2288	US	W 85%	<25 (ref), 25–29.9, ≥30	ABF	Cessation	-	2 months	86% (2 months)	OR	-	3
Hauff and Demerath (2012) [81]	239	US	C 85%	<25 (ref), ≥25.0	ABF	Cessation	-	6 months	76% (6 months)	HR	DM, AvBFP, PDB	3
Bartok (2012) [82]	672	US	W 91%	18.5–24.9 (ref), 25–29.9, ≥30	ABF	Cessation	-	6 months	51% (6 months)	HR	Edu, Mar, PDB, MRBI	5
Kronborg (2012) [84]	1442	Denmark	Danish	<27 (ref), 27–31.9, ≥32	ABF	Cessation	-	17 weeks	-	HR	Age, Duration of schooling	3
Branger (2012) [85]	239	France	C	<30 (ref), ≥30.0	ABF	Cessation	-	-	89% (1 month), 27% (6 months)	OR	Age, Decision of BF, P, O, BW, Confidence, Difficulty, Situation, Support of BF, DLM	4
Perrine (2012) [95]	1457	US	W 88%	<18.5, 18.5–24.9 (ref), 25.0–29.9, ≥30	EBF	Cessation	-	-	45% (3 months)	OR	Age, R/Eth, PTI ratio, Edu, P, Smk, NPFWIC, DLM, Mar, Intended duration of EBF, BFHI practices	6
Thompson (2013) [83]	1,161,949	US	H 32%, B 18%	<18.5, 18.5–24.9 (ref), 25–29.9, ≥30	ABF	Initiation	Immediate postpartum	-	79% (immediate postpartum)	OR	Age, R/Eth, Edu, MH, PNC, BW, InfSex, P, GW, Birth year	5
Sipsma (2013) [86]	225	US	B 40%, H 42%, O 18%	<18.5, 18.5–24.9 (ref), 25–29.9, ≥30	ABF/EBF	Initiation	-	-	71% (ABF initiated), 16% (6 months)	OR	Age, R/Eth, Edu, Em, NPFWIC, School status, FB	3
Brown (2013) [87]	4533	Canada	Can	<18.5, 18.5–24.9 (ref), 25–29.9, ≥30	EBF	Cessation	-	6 months	10% (6 months)	HR	E, Inc, Mar, Smk, P, FAS, Int, EBC	5
Gubler (2013) [88]	1893	Swiss	C 77%	<18.5, 18.5–24.9 (ref), 25–29.9, ≥30	EBF	Initiation	Discharge	-	68% (discharge)	OR	P, GWG, BW, DLM, Anesthesia, Hb <9.5 g/dL, DSH, SN, EBC, First suckling and rooming-in	3
Visram (2013) [89]	22,131	Canada	Can	18.5–24.9 (ref), 25–29.9, ≥30	EBF	Initiation	Discharge	-	68% (discharge)	OR	Age, NFI, NEL, AOR, P, First trimester visit, PNC, HCP, SFGA, DM, DLM, NICU admission	5
Zhu (2013) [116]	1602	China	Asian	<18.3,18.4–19.6 (ref), 19.7–21.2, ≥21.3	ABF	Cessation	-	2 months	88% (2 months)	OR	Age, Edu, Inc, SS, LE, GWG, GW, DLM, BF on day 1, OL	6
Hayes (2014) [90]	8508	US	Multiethnicities	<18.5, 18.5–24.9 (ref), 25–29.9, ≥30	EBF	Cessation	-	8 weeks	37% (8 weeks)	RR	R/Eth, Age, DLM, RTW, Dep	6
Vurbic (2014) [91]	300	US	C	<25 (ref), ≥25.0	ABF	Cessation	-	24 weeks?	70% (abstainer), 43% (smoker)	OR	Edu, Mar	3
Stuebe (2014) [92]	2335	US	W 81%	<18.5, 18.5–24.9 (ref), 25–29.9, ≥30	ABF	Cessation	-	12 months	18.6% (12 months)	OR	NPFWIC, Mar, R/Eth, Age, P, Edu	5
Hauff (2014) [93]	2423	US	W 85%	18.5–24.9 (ref), 25–29.9, ≥30	ABF/EBF	Cessation	-	-	93% (ever breastfed)	HR	R/Eth, planned RTW, PBE, PIFI, Mar, Smk, GW, PDB, Edu	3
Dashti (2014) [117]	345	Kuwait	Arabian	<25 (ref), 25–29.9, ≥30	EBF	Cessation	-	6 months	39% (6 months)	OR	-	3
Cox (2015) [94]	489	US	C	<30 (ref), ≥30	ABF/EBF	Cessation	-	26 and 52 weeks	ABF 76% (4 months), EBF 5.7% (6 months)	HR	Age, MPFFP, InfSex, BW, P, ADM, CA, PCAP, Edu, DLM, Inc, MS, O (father and mother), EBC, Room, PBE, TTRC, Smk	5
Smith (2015) [96]	874	Ireland	74% born in Ireland	<18.5, 18.5–24.9 (ref), 25–29.9, 30.0–40.5	ABF/EBF	Initiation/Cessation	Discharge	2 and 6 months	EBF 43% (discharge), 0.7% (6 months)	OR	GW, DLM, BW, ADM, Duration of hospital stay, Paternal BMI, Smk, Edu, O, Maternal and paternal R/Eth, Mar, Age	8
Winkvist (2015) [97]	16,922 who gained weight within the IOM recommendation	Norway	Residents in Norway	<18.5, 18.5–24.9 (ref), 25–29.9, ≥30	ABF/EBF	Initiation/Cessation	1 week	4 and 6 months	ABF 81% (6 months), EBF 14% (6 months)	OR	Inc, Edu, Age, Smk, P, DM, DLM	8
Forster (2015) [113]	924	Australia	Aus born 68%	<30 (ref), ≥30	ABF	Cessation	-	6 months	68% (6 months)	OR	-	3
Verret-Chalifour (2015) [98]	6592	Canada	C 97%	<18.5, 18.5–24.9 (ref), 25–29.9, ≥30	ABF	Initiation	After delivery	-	87% (in hospital)	RR	Place, Year of delivery, Age, Edu, Mar, P, Eth, Inc, Smk, Alc, History of drug use, BH, PBE, GW, HTN, InfSex, BW, APG	6
Masho (2015) [99]	95,141	US	W 60%	<18.5, 18.5–24.9 (ref), 25–29.9, ≥30	BF	Initiation	After delivery	-	83%	OR	Edu, NPFWIC, DOU, Age, PNC, Dep, Inc, Smk, PV	4
Lindau (2015) [100]	605	Italy	C	<18.5, 18.5–24.9 (ref), 25–29.9, ≥30	EBF	Cessation	-	6 months	16% (4 months)	OR	-	3
Castillo (2016) [119]	4011	Brazil	W 62%	<18.5, 18.5–24.9 (ref), 25–29.9, ≥30	EBF	Cessation	-	3 months	27% (3 months)	HR	Age, Edu, P, DM, HTN, Alc, InfSex, GW, BW, DLM	7
Kair (2016) [101]	2530 who had late preterm infants	US	W 95%	Underweight, normal, overweight, obese	ABF	Cessation	-	6 months	-	OR	Age, Edu, Mar, Insurance, P, DLM, NICU, BW, AOR	7
Logan (2016) [102]	856	Germany	German nationality 85%	<18.5, 18.5–24.9 (ref), 25–29.9, ≥30	ABF	Cessation	-	4 and 6 weeks; 6 months	ABF 67% (6 months)	HR	Edu, Alc, Smk, DLM, RTW	4
Tao (2017) [115]	3196	China	Asian	<18.5, 18.5–23.9 (ref), 24–27.9, ≥28	ABF/EBF	Cessation	-	1, 3 and 6 months	EBF 44% (1 month), 51.6% (3 months), 11% (6 months)	HR/RR	Age, GW, BW, DLM, InfSex, GWG, Eth, Edu, Inc, P, Smk, Alc, HTN, DM	8
de Jersey (2017) [114]	329	Australia	Aus born 72%	<24.9 (ref), ≥25	EBF	Initiation/Cessation	-	4 months	82% (discharge), 40% (4 months)	OR	Edu, ROC, DLM,	3
Tehranian (2017) [118]	593	Iran	Arabian	<24.9 (ref), ≥25	ABF/EBF	Initiation/Cessation	-	6 months	EBF 93% (1 month)	OR	Age, Edu, DLM, M’s opinion, O	5
Haile (2017) [104]	2026	US	W 85%	<18.5,18.5–24.9 (ref), 25–29.9, ≥30	EBF	Initiation/Cessation	-	3 months	75% (discharge), 28% (3 months)	OR	Age, Eth, Edu, MS, PTI ratio, DM, GWG, PIFI, DLM, Smk, GW, ADM, BFHI	6
Wallenborn (2017) [105]	34,854	US	W 60%	<18.5, 18.5–24.9 (ref), 25–29.9, ≥30	BF	Initiation	-	-	75%	OR	Age, Edu, R/Eth, Inc, Insurance, PNC, MedS, WIC, Stressors	4
Bjørset (2018) [103]	700	Norway	Nor 87%	<18.5, 18.5–24.9(ref), 25–29.9, ≥30	ABF/EBF	Cessation		4 and 5 months	ABF 81% (5 months), EBF 52.7% (4 months)	OR	Age, Edu, Mar, P, Smk, APUE, Degree of urbanization, APUE, DOU	4
Ramji (2018) [107]	12,422	Canada	Can	18.5–24.9 (ref), 25–29.9, 30–39.9, 40–49.9	ABF	Initiation	Discharge	-	68% (discharge)	OR	Age, P, POC, Em, Edu, Smk, DM, HTN, Anesthesia	4
Marshall (2019) [106]	140	US	W 84%	<24.9 (ref), ≥25	EBF	Initiation/Cessation	6 weeks	6 months	80% (6 weeks), 66% (6 months)	OR	Age, GWG, Smk, PDB	5
Nomura (2019) [16]	6125	Japan	Asian	<18.5, 18.5–24.9(ref), 25–29.9, ≥30	EBF	Initiation	1 month	-	83% (discharge)	OR	Age, P, DLM, Alc, smk, GW, a light for date infant, maternal–child separation	6

Abbreviations: ABF, Any breastfeeding; ADM, admission to special care nursery; Alc, Alcohol; AOD, Anxiety or depression problem; AOR, Area of Residence; APG, Apgar score; APUE, ability to pay unforeseen expenses of 3000 NOK; Aus, Australian; AvBFP, avoidance breastfeeding in public; B, Black; BF in Day1, Breastfeeding frequency in Day1; BF, breastfeeding; BFHI, Baby-Friendly Hospital Initiative; BH, Breast History; BMI, Body Mass Index; BW, Birth weight; C, Caucasian; Can, Canadian; Dep, Self-reported Depressive symptoms; Desire, Desire to breastfeed; DLM, Delivery method; DM, Diabetes Mellitus including gestational diabetes; DML, Duration of maternity leave; DOU, degree of urbanization; DSH, duration of stay in hospital; Em, Employment; EBC, Early breast contact; EBF, Exclusive breastfeeding including Full breastfeeding; Edu, Education; FAS, Folic acid supplementation; FB, first baby; FP, feeding preference; FTIPC, first trimester initiation of prenatal care; GW, Gestational Week; GWG, Gestational Weight Gain; H, Hispanic; HC, Health concession card holder; HCP, HealthCare Provider; HR, hazard ratio; HTN, Hypertension including gestational hypertension; IFS, infant feeding score; InfSex, infant Sex; Inc, income; Int, intention to BF; Jpn, Japan; LE, Life event in third trimester; M’s opinion, mothers’ opinion of breastfeeding; Mar, Marital Status; MedS, Medicaid status during pregnancy; MH, Maternal Health; MOI, Model of institution; mos, months; MPFFP, mother’s perception of father’s feeding preference; MRBI, maternal rating of breastfeeding importance; N/A, Not available; NEL, Neighborhood education level; NFI, neighborhood family income; NICU, Newborn Intensive Care Unit; Nor, Norwegian; NPFWIC, Nutrition program for Women, Infant and Child; O, Occupation; OL, Onset of lactation; OR, odds ratio; P, Parity, PA, Physical activity; PB, Pregnancy Problem; PBE, Previous Breastfeeding Experience; PCAP, Prenatal Care Assistance Program; PDB, planned duration of breastfeeding; PH, Past history of being breastfed; PIFI, Prenatal infant-feeding Intention; PNC, prenatal care; POC, Presence of company including a spouse or partner during early pregnancy; PS, Poverty Status; TI ratio, poverty-to-income ratio; PV, Partner Violence; R/Eth, Race/Ethnicity; ref, reference; RH, Received formula in hospital; ROB, Region of birth; Room, Room in hospital; RP, Relationship problems; RTW, Return to work/school; SC, Social security; SED, level of socio-economic disadvantage of the geographical location of the child’s household; SFGA, small for gestational age; Smk, Smoking; SN, sore nipple; SS, Social Support; timing, timing of infant feeding decision; UK, United Kingdom; US, United States; W, White; wks, weeks;WIC, Women, Infant, and Children Food and Nutrition Service/Program3.3. Quality of 57 Selected Studies.

**Table 2 nutrients-12-02684-t002:** Pooled risk of breastfeeding non-initiation with 95% confidence interval (CI), heterogeneity and publication bias.

Breastfeeding Initiation	*n*	Pooled Risk (95% CI)	Tau^2^	I^2^	*p* for Q	Meta-Regression	Begg’s Test
						*p*-Value	
**Underweight vs. normal weight**							
Overall	14	1.03 (0.85–1.23)	0.077	88	<0.001	-	0.344
Subgroup analyses							
Quality score							
6–	6	0.97 (0.71–1.34)	0.097	79	0.247	0.537	
−5	8	1.06 (0.84–1.34)	0.073	88	0.003	ref	
Overweight vs. normal weight							
Overall	29	1.33 (1.15–1.54)	0.115	98	<0.001	-	<0.001
Subgroup analyses							
Quality score							
6–	9	1.27 (1.01–1.60)	0.093	79	0.003	0.674	
−5	20	1.39 (1.15–1.67)	0.126	99	<0.001	ref	
Obese vs. normal weight							
Overall	26	1.61 (1.33–1.95)	0.195	99	<0.001	-	0.013
Subgroup analyses							
Quality score							
6–	9	1.82 (1.28–2.58)	0.232	87	<0.001	0.379	
−5	17	1.51 (1.20–1.90)	0.18	99	<0.001	ref	

**Table 3 nutrients-12-02684-t003:** Pooled risk of underweight, overweight and obesity vs. normal weight on ABF cessation.

Breastfeeding Cessation	*n*	Pooled Risk (95% CI)	Tau^2^	I^2^ (%)	*p* for Q	Meta-Regression	Begg’s Test
						*p*-Value	
**Underweight vs. normal weight**							
Hazard ratio	4	1.28 (0.71–2.31)	0.317	95	0.024	-	0.013
Odds ratio or risk ratio	10	0.91 (0.64–1.28)	0.188	86	0.002	-	0.214
Quality score							
6–	7	0.87 (0.54–1.39)	0.299	93	<0.001	0.915	
−5	3	0.92 (0.58–1.44)	0.005	2	0.854	ref	
Overweight vs. normal weight							
Hazard ratio	10	1.16 (1.07–1.25)	0.003	23	0.913	-	0.888
Quality score							
6–	3	1.14 (1.03–1.26)	0.001	14	0.620	0.626	
−5	7	1.18 (1.07–1.31)	0.004	19	0.836	ref	
Odds ratio or risk ratio	16	1.17 (1.00–1.38)	0.073	89	<0.001	-	0.306
Quality score							
6–	9	1.28 (1.07–1.53)	0.048	86	0.001	0.125	
−5	7	1.04 (0.79–1.38)	0.099	81	0.001	ref	
Obese vs. normal weight							
Hazard ratio	10	1.43 (1.22–1.69)	0.044	68	0.081		0.010
Quality score							
6–	2	1.35 (1.10–1.65)	<0.001	1	0.694	0.695	
−5	8	1.48 (1.21–1.81)	0.056	73	0.033	ref	
Odds ratio or risk ratio	18	1.47 (1.16–1.86)	0.200	93	<0.001		0.371
Quality score							
6–	7	1.71 (1.34–2.18)	0.063	85	<0.001	0.185	
−5	11	1.32 (0.94–1.84)	0.265	90	<0.001	ref	

**Table 4 nutrients-12-02684-t004:** Pooled risk of underweight, overweight and obesity vs. normal weight on EBF cessation.

Breastfeeding Cessation	*n*	Pooled Risk (95% CI)	Tau^2^	I^2^ (%)	*p* for Q	Meta-Regression	Begg’s Test
*p*-Value
Underweight vs. normal weight							
Hazard ratio	3	1.26 (0.69–2.31)	0.238	88	0.026	-	0.029
Odds ratio or risk ratio	9	1.02 (0.87–1.21)	0.028	72	0.535	-	0.341
Quality score							
6–	7	1.04 (0.87–1.22)	0.028	77	0.38	0.625	
−5	2	0.83 (0.39–1.76)	0.019	5	0.535	ref	
Overweight vs. normal weight							
Hazard ratio	6	1.13 (1.00–1.28)	0.013	57	0.264	-	0.689
Quality score							
6–	2	1.06 (0.84–1.34)	0.019	65	0.08	0.54	
−5	4	1.16 (1.00–1.35)	0.010	47	0.356	ref	
Odds ratio or risk ratio	12	1.37 (1.10–1.72)	0.127	96	<0.001	-	<0.001
Quality score							
6–	7	1.15 (1.08–1.24)	0.003	48	0.059	0.065	
−5	5	1.83 (1.08–3.11)	0.295	86	0.001	ref	
Obese vs. normal weight							
Hazard ratio	7	1.45 (1.27–1.65)	0.012	44	0.455		0.450
Quality score							
6–	2	1.38 (1.04–1.82)	0.014	28	0.295	0.926	
−5	5	1.46 (1.27–1.69)	0.011	44	0.328	ref	
Odds ratio or risk ratio	11	1.32 (0.97–1.79)	0.205	96	<0.001		0.706
Quality score							
6–	7	1.43 (1.19–1.71)	0.037	86	0.011	0.147	
−5	4	0.93 (0.44–1.98)	0.457	86	0.001	ref

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
