# Peer review of "Maternal Body Mass Index and Breastfeeding Non-Initiation and Cessation: A Quantitative Review of the Literature"

_nutrients, 2020, doi:10.3390/nu12092684_

Round 1
Reviewer 1 Report
This is an interesting systematic review and meta-analysis that assesses the maternal body mass index and its association with the initiation and continuation of breastfeeding. The benefits of breastfeeding are well-established; however, a critical appraisal of the current literature would also highlight the importance of a normal BMI for the breastfeeding. In addition, there are several guidelines regarding the appropriate BMI, as well as the appropriate weight gain during pregnancy, in order to acquire uneventful outcomes for both the mother and the neonates.
Regarding this meta-analysis, my main concern is that a similar one was published in 2019.
Hence, I recommend that the authors create a table to indicate the differences (studies included, exclusion criteria, reasons for not including some studies) with the previous meta-analyses. Then, if the latter identifies a significant contribution to the literature, the current meta-analysis could be accepted for publication.
Author Response
We would like to express our deep appreciation for the productive and thoughtful comments from the reviewers.
# Regarding this meta-analysis, my main concern is that a similar one was published in 2019. Hence, I recommend that the authors create a table to indicate the differences (studies included, exclusion criteria, reasons for not including some studies) with the previous meta-analyses. Then, if the latter identifies a significant contribution to the literature, the current meta-analysis could be accepted for publication.
→Referring to the comments from the reviewer 1, we have created the attached table. TABLE 1. Research methodology that differentiate our study from the study conducted by Huang, et al.
The advantage of our study over Huang, et al. is that first, the numbers of studies included in meta-analyses are much larger in our study than in Huang, et al. We included 57 studies (Initiation n=29, ABF cessation n=31, EBF cessation n=20) vs. 30 in Huang et al. (Initiation n=21, ABF cessation n=15, EBF cessation n=17). Although the search strategy and inclusion/exclusion criteria are different between ours and Huang, et al., we identified 29 studies of 30 studies that Huang, et al. selected in meta-analyses and included 24 studies in our meta-analyses according to our inclusion criteria. One of the clearest differences is that Huang, et al. excluded cross-sectional studies. However, the exclusion of those non-longitudinal studies may overlook the true association of interest because most studies that focused on breastfeeding initiation are cross-sectional studies. Second, we performed comprehensive literature review by including a larger number of literatures (i.e., 131 in our study vs. 54 in Huang, et al.) and extended discussion by not limited to the studies included in meta-analyses but including excluded studies in Supplementary Table 2, “Characteristics of excluded studies that investigated BMI as a continuous variable”, for example. The description of those excluded studies from meta-analyses is of importance to reproduce a systematic review for a third parson. Third, Huang, et al. combined odds ratios with hazard ratios, but, because these two risk ratios have different epidemiological concept, we performed pooled analyses according to odds or hazard ratios (We have discussed with our statistician, Prof. Kengo Nagashima and agreed with this important procedure). Thus, our sophisticated analyses decreased the levels of heterogeneity in overweight and obese which made it clearer to present true association of interest that overweight and obese women are associated with an increased risk of breastfeeding non-initiation or cessation. Fourth, although Huang, et al. reported that underweight mothers had a negative impact on BF initiation, we did not observe such a negative impact on underweight women. Instead, we see substantial heterogeneity across studies in underweight women and confirmed that such substantial heterogeneity was also observed in the study conducted by Huang, et al. To summarize, the effect of underweight on breastfeeding non-initiation or cessation risk still require evidence accumulation. Based on the difference mentioned above, we updated our revised manuscript to enhance the difference of study methodology between ours and Huang, et al. in Discussion, line 371-393.
There are two previous meta-analysis studies [14,120] that investigated an association between maternal weight status and BF non-initiation and cessation and agree that obesity may be associated with an increased risk of BF non-initiation or cessation compared to normal weight mothers. However, their study hypotheses differ from ours; one study [120] not only included maternal weight status, but diet and supplement use as determinants of breastfeeding and complementary feeding; the other [14] investigated BF cessation associated with maternal BMI and gestational weight gain. The study conducted by Huang et al. [14] is similar to ours, but does not include cross-sectional studies, resulting in only 30 studies. The exclusion of these studies may overlook the true association of interest because most studies that focused on breastfeeding initiation are cross-sectional studies. Hence, the advantages of our study are that, first, the numbers of studies included in meta-analyses are much larger in our study than in Huang et al. We included 57 studies (Initiation n=29, ABF cessation n=31, EBF cessation n=20) vs. 30 in Huang et al. (Initiation n=21, ABF cessation n=15, EBF cessation n=17). Second, we also performed a comprehensive literature review including a large number of references (i.e., 130 in our study vs. 54 in Huang et al.). For example, our study encompassed an extended literature search not limited to only the studies included in the meta-analyses, but incorporating those studies that were excluded into our supplementary table (e.g., Supplementary Table 2 “Characteristics of excluded studies that investigated BMI as a continuous variable”). The description of studies excluded from the meta-analyses is important to reproduce a systematic review for a third person. Second, by performing pooled analyses according to odds or hazard ratios, heterogeneity of some meta-analyses is less observed which makes it clearer to present the true association of interest. The discrepancy is that Huang et al. reported underweight mothers to be less likely to initiate BF compared to normal weight mothers (n=14, RR 1.28, 95% CI:1.11-1.48). However, we did not observe such a negative impact among underweight women; both studies observed substantial heterogeneity across the studies included in the meta-analyses. In this regard, the effect of the mother being underweight on breastfeeding still requires further evidence.

Reviewer 2 Report
Nomura and co-author investigate which maternal BMI categories are associated with the initiation or continuation of breastfeeding, based on quantitative reviews. Their quantitative review suggests that overweight and obese mothers are more likely to fail in breastfeeding initiation and continuation. This review contributes to a better understanding of the role of maternal body condition on breastfeeding initiation and continuation. I am concerned about some points.
- Why do the authors selecte MEDLINE and CINAHL databases for reference collection?
- Do other factors influence the results, such as region and climate?
- Line 70:1980-2019, Line 194:1966-2019? Please confirm the date.
- In table 1, the number of references is not correct.
- Did the authors adjust the data from references? Different papers may use various method to measure the same index.
Author Response
We would like to express our deep appreciation for the productive and thoughtful comments from the reviewers. The following are our point-to-point responses to your comments. The revised sentences in the manuscript are highlighted with track-changes embedded in Word.
#Why do the authors select MEDLINE and CINAHL databases for reference collection?
→We selected these two databases because MEDLINE and CINAHL are two major databases to be considered for literature review in an area of Public Health. A previous study published in Syst Rev (Bramer, et al. 2017;6(1):245. doi: 10.1186/s13643-017-0644-y.) recommends at least Embase, MEDLINE, Web of Science, and Google Scholar as a minimum requirement to guarantee adequate and efficient coverage while other specialized databases, such as CINAHL or PsycINFO, to add unique references to some reviews in a specific field (CINAHL for Public Health).
#Do other factors influence the results, such as region and climate?
→There are several factors that influence breastfeeding as we mentioned in the Introduction. These include biological factors such as maternal characteristics or delivery outcome, social environmental factors, and even the perception of infant feeding preference of the parents and family but none of the previous literatures had ever reported region or climate as a determinant. Nevertheless, this is an interesting research question that we may want to try in the future. We thank the reviewer for such an interesting insight.
#Line 70:1980-2019, Line 194:1966-2019? Please confirm the date.
→We thank the reviewer for the notice. We have included the date as follows.
2.1. Literature search
A literature search was performed of the MEDLINE and CINAHL databases from January 1946 and January 1981, respectively, through June 8th 2019 by using key terms
#In table 1, the number of references is not correct.
→We thank the reviewer nor this note. We deleted a row of “No” because these numbers do not indicate each reference number.
#Did the authors adjust the data from references? Different papers may use various method to measure the same index.
→We thank the reviewer for the concern. Please refer to a row “Adjusted variables to estimate the risk of breastfeeding non-initiation or cessation“ in Table 1. We described whether the data are adjusted or not and then used adjusted data if they are available, which are reflected in 2.4.Outcome of breastfeeding.
2.4.Outcome of breastfeeding
The outcome of interest in our meta-analyses was any or exclusive breastfeeding (ABF and EBF, respectively) in relation to non-initiation or cessation. According to the definition proposed by the World Health Organization [31], children were considered to be exclusively breastfed if they were given no other food or drink, or added nonhuman milk or infant formula, except breast milk. ABF was considered if the child received liquids like water-based drinks, fruit juice, and ritual fluids, other than breast milk [31], or if the study did not clearly describe exclusive or any. We collected either ORs or HRs of BF non-initiation or cessation and if the study provided risks of BF initiation or continuation, we recalculated the risk of BF non-initiation or cessation by taking the inverse of the risk of the BF status to align the direction of the risk. To estimate the pooled effect size of the risk measures, we used risk ratios in multivariable analyses rather than unadjusted models.

Round 2
Reviewer 1 Report
I would like to thank you for the prompt revision. I recommend accept of the article in its present form.
Reviewer 2 Report
Agree to publish this MS in Nutrients.